# Rodent Models of Dilated Cardiomyopathy and Heart Failure for Translational Investigations and Therapeutic Discovery

**DOI:** 10.3390/ijms24043162

**Published:** 2023-02-05

**Authors:** Matteo Ponzoni, John G. Coles, Jason T. Maynes

**Affiliations:** 1Division of Cardiovascular Surgery, The Hospital for Sick Children, Toronto, ON M5G 1X8, Canada; 2Program in Translational Medicine, SickKids Research Institute, Toronto, ON M5G 0A4, Canada; 3Department of Anesthesia and Pain Medicine, The Hospital for Sick Children, Toronto, ON M5G 1X8, Canada; 4Program in Molecular Medicine, SickKids Research Institute, Toronto, ON M5G 0A4, Canada; 5Department of Anesthesiology and Pain Medicine, University of Toronto, Toronto, ON M5G 1E2, Canada

**Keywords:** rodent model, dilated cardiomyopathy, heart failure, review

## Abstract

Even with modern therapy, patients with heart failure only have a 50% five-year survival rate. To improve the development of new therapeutic strategies, preclinical models of disease are needed to properly emulate the human condition. Determining the most appropriate model represents the first key step for reliable and translatable experimental research. Rodent models of heart failure provide a strategic compromise between human in vivo similarity and the ability to perform a larger number of experiments and explore many therapeutic candidates. We herein review the currently available rodent models of heart failure, summarizing their physiopathological basis, the timeline of the development of ventricular failure, and their specific clinical features. In order to facilitate the future planning of investigations in the field of heart failure, a detailed overview of the advantages and possible drawbacks of each model is provided.

## 1. Introduction

Thanks to the advances in medical therapies for heart failure (HF), as well as the persistent improvements in cardiovascular surgical techniques and devices technologies, many previously fatal cardiac diseases are now chronically managed with satisfactory medium- and long-term prognosis. However, once the cardiac performance is severely affected, a progressively declining syndrome is established, whose definitive fate is often end-stage organ failure [1]. Demographic projections estimate that by the next decade one out of every thirty-three people in the US will be affected by HF [2]. In the highly complex population of pediatric patients with structural heart defects, the number of patients palliated with the Fontan procedure is expected to double in the next 20 years [3]. Currently, more than 90% of children with congenital heart defects survive to adulthood [4], indicating a substantial increase in demand for HF-related services as this population ages into adulthood with expected evolving cardiac dysfunction.

Current treatment options for end-stage HF entail organ replacement with human grafts, artificial devices, or xenografts. However, none these strategies are definitively curative (incorporating graft failure) and are limited by the availability of donors [5], chronic immune rejection [6,7], high rates of graft vascular disease, and abnormal activation of coagulation cascades and platelets at the blood–device interface [8,9]. Conversely, regenerative medicine techniques may represent a new frontier in the management of end-stage HF. Cardiac progenitor stem cells and derived proteins [10,11], molecularly targeted drugs [12], and reinvented surgical procedures [13,14] aim at stimulating the intrinsic repair ability of the human heart [15,16]. Genome editing technologies are emerging as potential therapeutic strategies able to address specific causative monogenetic disorders associated with the development of cardiomyopathy and HF. Through base editing, the expression of key proteins that are dysregulated in rare genetic forms of dilated cardiomyopathies (DCM), such as dystrophin in Duchenne muscular dystrophy [17] and Rbm20 [18], can be restored, representing a promising therapeutic concept.

To test novel putative therapeutic approaches to HF in a controlled manner, viable animal models of HF are paramount. Due to an advantageous compromise between adequate body size and low housing and maintenance costs, rodent models of HF have been extensively used by basic and translational scientists for decades [19,20,21]. Particularly, rats offer sufficient dimensions to perform cardiac surgical procedures and invasive hemodynamic measurements safely, shortening learning curves for operators. Moreover, they expedite advanced imaging measurements and provide a 10-fold greater myocardial mass for subsequent histopathological or molecular analyses, when compared to mice.

Many methods have been tested and validated to induce progressive HF and DCM in rats, mimicking the different etiologies of HF in humans. With the present review, we illustrate the available rat models of HF (Figure 1), highlighting their strengths and drawbacks. A comprehensive description of their mechanisms and timelines for developing HF is provided, as well as their documented or potential purposes for translational research and experimental surgery.

## 2. Rats and Mice as Animal Models of Heart Failure

Rats and mice share the benefits of small mammalian models, in terms of ease of handling and housing, short breeding cycle, and the number of recruitable animals to improve the statistical power of experiments. The small physical size reduces the costs of novel therapeutic agents and molecules, whose administration is usually calculated based on body weight. However, some differences apply between these two species, starting with the size of the animals. Despite a similar 2–3 years life span, adult (6–8 week-old) rats weigh 250 to 550 g, while adult mice are generally 10 times smaller (25–30 g) [22]. This aspect becomes particularly valuable in the setting of experimental surgery. Animal intubation and ventilation, access to anatomical structures of interest, and performing specific procedures are easier in larger rodents, shortening learning times for operators and increasing animal survival. Moreover, invasive micromanometer catheterization can be accomplished more safely in rats, and the spectrum of applicable noninvasive echocardiographic and magnetic resonance imaging techniques mimics the one available in the clinical setting [23,24]. Finally, a greater number of different postmortem analyses can be accomplished on the same rat, thanks to the larger size of tissues and higher intravascular blood volume.

Rodent cardiac physiology, contraction patterns, and energetics resemble the human equivalents. Rodent and human myocardial tissue share similar functions for many proteins, although rodent cardiomyocytes exhibit a predominance of alpha-myosin heavy chains (compared to beta-myosin heavy chains in humans), which are characterized by a rapid ATPase activity to facilitate the extremely high heart rate and the short cardiac cycle. Mice present a resting heart rate of 500 to 600 bpm, with 350 bpm for rats. Although still far from human values, the de-escalation from mice to rats in terms of cellular bioenergetics and mitochondrial efficiency contributes to translational value [25]. These differences should be taken into account particularly in the setting of preventative and reparative molecular therapies for HF since they could influence the likelihood of moving findings into human clinical practice [21]. An intermediate validation in large animal models is mandatory for this purpose (i.e., porcine).

The mechanisms of HF are often multifactorial, involving several risk factors or acquired conditions that induce or accelerate myocardial damage, with or without a predisposing genetic background. Small animal models are particularly useful in the study of specific risk factors that promote HF, avoiding the confounding effects of external variables present in the clinical setting. Single pathophysiological pathways can be investigated: myocardial ischemia, pressure and volume overload, toxins, diabetes mellitus, atherosclerotic disease, and several metabolic syndromes [20,26]. In this setting, rat models are broadly adopted to test therapeutic hypotheses and target specific sensitive pathways [19]. Moreover, rat models were the first to be used for the study of the combination of different HF mechanisms in the same animal [27].

The high degree of genetic similarity between rodents and humans allowed for the creation of relevant transgenic and knockout strains with an HF phenotype [21]. Easier manipulation of the mouse genome explains the extremely large number of transgenic mouse strains currently available for target analysis. Mouse models obtained from mutations in cardiac myosin light chain, lamin, troponin, and other extra or intrasarcomeric proteins have been addressed in previous reviews [28,29,30,31]. Conversely, genetic rat models of HF are infrequent [21]. The translation of molecular mediators resulting from genetically modulated rodents into human studies requires a cautious approach. Genetic mutations that cause idiopathic DCM in humans are mostly undiscovered [32]. Although many proteins share similar functions, their expression levels and the final organ and biological effects can differ substantially between rodents and larger mammals and humans [19,21]. As an example, dystrophin-deficient mice exhibit a normal life span and mild cardiomyopathy, which is in contrast with the human phenotype of Duchenne muscular dystrophy [33]. Testing genetic hypotheses in large models is therefore recommended before human trials.

## 3. Rat Models: General Considerations

The ideal animal model should be able to reproduce the typical echocardiographic, histological, and clinical features of the desired type of HF. Reflecting the higher prevalence of left-sided HF in the general population, left ventricular (LV) dysfunction models leading to HF and eventually DCM are more widely utilized. Human DCM is defined as a spectrum of myocardial diseases which share ventricular dilatation and depressed contractility [34]. The key phenotype is characterized by a progressive LV dilatation, together with a ventricular shape transition from its original ellipsoid shape to a more spherical one, wall thinning, and a global reduction in contractility, which is revealed by a decrease in stroke volume, cardiac index, and increased strain parameters [32,35]. These features differentiate DCM from other cardiomyopathies, such as hypertrophic cardiomyopathy (where increased LV wall thickness and normal or even supranormal contractility is noted [36]), restrictive cardiomyopathy (in which ventricular chamber dimensions are reduced, impairing LV filling and creating a primary diastolic dysfunction [37]), and arrhythmogenic right ventricular cardiomyopathy (characterized by typical electrocardiographic anomalies and an often pathognomonic fibrous-fatty myocardial replacement [38]). The available animal models of hypertrophic, restrictive, and arrhythmogenic cardiomyopathies have been extensively reviewed in previous publications [39,40,41,42,43,44].

In DCM, associated diastolic dysfunction can occur, and the combination of increased LV filling pressures and ventricular dilatation often generates functional mitral regurgitation. Magnetic resonance imaging can detect areas of late gadolinium enhancement, which represent the process of diffuse fibrosis that is common to many DCM etiologies (the slower heart rate of the rat also makes cardiac MRI more feasible, although specialized instrumentation is still needed). Endomyocardial biopsy typically reveals morphological alterations in DCM: myocardial disarray, fibrosis, cell death, cardiomyocyte hypertrophy, scar formation, and inflammatory infiltration [45]. Importantly, this phenotype could characterize only the LV or present with biventricular involvement, depending on the etiology and severity of the disease. This aspect requires particular attention when choosing the most adequate animal model for the desired experimentation.

To date, a benchmark rat model of DCM is not available. Existing animal models are not able to respond to all of the above-mentioned requisites of the ideal model. The main reason lies in the fundamental difference in the development of human DCM versus the induction of experimental DCM. As previously stated, DCM in humans has a wide spectrum of causes, but in most of the cases a leading factor cannot be detected, often with evolution over significant time periods. Experimental DCM models rely mostly on a single pathogenic pathway with relatively rapid onset, which can consequently reproduce only some specific features of the DCM phenotype.

Recently, scientific interest has moved towards other variants and causes of HF, generating the need for new, representative animal models. Heart failure with preserved ejection has been recognized as an independent clinical entity with specific etiologies and pathogenic pathways involved [46]. Right ventricular (RV) failure is a considerable issue in patients who have survived surgical correction of congenital heart defects during infancy, as well as those children and adolescents palliated with a Fontan circulation utilizing a systemic RV [47]. In this view, novel rodent models of HF not progressing to DCM or specifically involving the RV have been developed. To summarize, there are seven main detrimental stimuli which can induce HF in preclinical models: ischemic injury; pressure overload; volume overload; drug toxicity; autoimmunity; rapid pacing; and genetic mutations (Figure 1 and Table 1). All these models will be discussed in the following sections.

## 4. Ischemic Injury Models

Ischemic injury in rats has been induced in different ways: subcutaneous injections of isoproterenol [49], direct damage using electrocautery [50], arterial ligation, or cryogenic damage with a metal probe cooled in liquid nitrogen [51]. In 1979, Pfeffer et al. introduced the left anterior descending (LAD) artery ligation model [55], which now has become the preferred ischemic model. A direct correlation exists between the infarct size and the severity of LV dilatation and contractile function impairment, which ranges from completely preserved if the scar involves <30% of the LV circumference to congestive HF if >46% [55]. However, the predictability of the infarcted myocardial area after LAD ligation is less consistent and varies depending on the level where the suture is placed and anatomical variation [56]. Coronary anatomy in rats presents significant variability between animals: the single septal branch may originate from the proximal left coronary artery in 60% of cases and from the proximal right coronary in the remaining 40%; the circumflex artery branches distally from a long left main coronary artery in 66% of animals, while it arises more proximally from a short left main coronary artery or the main septal branch in 34% [56]. As a result, a distal LAD ligation always creates an infarction only in the LV anterior wall, while a proximal ligation (just below the left atrial appendage) can create a wider anterolateral infarction (64% of cases) or an only anterior infarction (36%). Standardized surgical protocols are now widely available to guide the investigator [53,54], helping to reduce model animal-to-animal variability.

An alternative ischemic model consists of ligation of the circumflex artery, which is demonstrated to produce a significant infarct zone (40% of LV diameter) [52]. However, this procedure is less validated and, being technically more demanding, should be adopted as a secondary option to the LAD ligation model.

A general concern regarding the ischemic rat model of HF is that the LAD ligation procedure is usually performed in young (4–8-week-old) rats, while ischemic diseases affect an older and multidiseased population. Since the regenerative potential of the human and mammalian heart is age dependent [15,134], and the causes of idiopathic DCM rely only marginally on an ischemic substrate, the transition from the LAD ligation model to clinical practice necessitates careful verifications in this specific context. On the other hand, the leading cause of congestive HF in humans remains coronary arterial disease, which the LAD ligation model clearly exemplifies. Thanks to the advancement of emergency care, most of the patients that present with an acute myocardial infarction with ST-elevation can now benefit from a prompt revascularization strategy [135]. Despite successful revascularization, up to 20% of patients surviving an ST-elevation myocardial infarction are hospitalized with HF in the first year after the event [136]. After revascularization, the underlying pathophysiological basis of myocardial damage switches from irreversible ischemia and cell necrosis to transient ischemia and ischemia-reperfusion injury. To account for these mechanisms, the LAD ligation procedure has evolved from a permanent ligation to a temporary (typically 30 min) ligation, followed by controlled reperfusion [57]. This approach allowed for the investigation of molecular pathways involved in the protective role of the ischemic preconditioning process [58]. Moreover, Petters et al. proposed an innovative model of transient LAD ligation followed by temporary (60 min) cardiopulmonary bypass support [59]. This model ingeniously mimics advanced extracorporeal membrane oxygenation or ventricular assist device protection during high-risk percutaneous or surgical coronary interventions and cardiogenic shock.

With the LAD or circumflex ligation models, it is clear that the subsequent impairment of myocardial performance will be limited to the LV. Although most of the reports using isoproterenol-induced HF describe changes in the LV, parallel hypertrophy in the RV has been observed, supporting a biventricular effect of this drug [49]. Electrocautery or cryogenic-based direct damage to the myocardium can theoretically be applied on every surface of the heart, potentially inducing isolated RV, isolated LV, or biventricular failure. While these methods generate predictable and controllable damage, these models are least similar to human physiology and are not preferred.

Despite the mentioned limitations of the ischemic rodent model, it is indisputable that this model represents a primary reference platform for the study of ischemic HF. As an example of translational ability, ischemic injury models have permitted the study of the reverse remodeling properties of angiotensin-converting enzyme inhibitors and angiotensin II receptor antagonists on LV volumes and performance [19], which now constitute the mainstays of medical therapy for HF [46]. Table 1 summarizes the advantages and drawbacks of each ischemic model of DCM.

## 5. Pressure Overload Models

Pressure overload can be generated in rats using surgical or non-surgical methods (Table 1). Surgical techniques usually entail an abrupt augmentation of the ventricular afterload by placing a tight band around the aorta or the pulmonary artery (PA). 

Aortic banding can be achieved at different levels. In the original procedure described by Rockman et al. in a mouse model [60], a suture is placed around the transverse aortic arch and tightened against a 27 G needle. The same technique has been implemented in rats (using an 18–22 G needle) and modified by applying a more proximal banding of the ascending aorta [62,63], which mimics the pathophysiological features of LV failure induced by severe aortic stenosis. Finally, the abdominal aorta can be banded to investigate the mechanisms of ventricular remodeling by creating a slower-developing hypertensive HF profile [68].

Tightening of the aorta or PA can be accomplished using a suture tied against a needle, with the needle rapidly removed to restore antegrade blood flow. Given the need for complete vessel mobilization and the related risk of fatal bleedings, alternative techniques have been proposed, such as half-closed surgical clips [69], which generate even higher pressure gradients across the PA with minimal mobilization of the vessel, or O-rings of predetermined diameters, which guarantee reproducible grades of aortic banding in mice [61].

Using these surgical approaches, delineated RV failure can be achieved 7 weeks from PA banding, as demonstrated by a reduced ejection fraction and cardiac output and severely dilated right chambers [70], with temporal sex-related differences in developing the HF phenotype (observed less contractile and diastolic dysfunction and fibrosis in females) [71]. The grade of PA banding contributes directly to the timing of the transition to a decompensated RV response, which can start manifesting after 1–3 weeks from the procedure in case of severe PA constriction [70,72]. On the other hand, aortic banding in rats produces an initial compensated hypertrophic response, with preserved ejection fraction and LV diameters 4 weeks after banding [64], which can last until 18–20 weeks before LV systo-diastolic failure occurs [62,65,66]. Interestingly, this adaptation to pressure overload is generally absent in mice, which develop very early LV failure and DCM [64]. Understanding the timing of progression from hypertrophy to cardiac decompensation in rat models is mandatory to guide the interventional and analytical planning of experiments.

Although technically challenging and with not insignificant mortality rates [66], rats can undergo aortic de-banding to investigate the antiremodeling effects of LV unloading which are accomplished by aortic valve replacement or ventricular assist devices in the clinical setting. After 6 to 9 weeks of aortic banding, LV unloading (i.e., de-banding) promotes the regression of hypertrophy and the recovery of diastolic function [73], supported by the restoration of mitochondrial energetics [74] and a reduction of pro-fibrotic factors [67]. Interestingly, aortic banding triggers significant systo-diastolic impairment and activation of prohypertrophic and profibrotic pathways also in the RV, which persist partially altered even after de-banding [63]. 

Surgical hypertensive models have been utilized to test reverse remodeling pharmacological strategies to treat HF, such as angiotensin-converting enzyme inhibitors and angiotensin II receptor antagonists [12,68,137] and, more recently, to guide targeted anti-fibrotic therapeutic molecules [67]. Interesting insights into the most appropriate timing to plan aortic valve replacement in the setting of asymptomatic aortic stenosis may be derived from these models in the future.

Non-surgical hypertensive models of LV failure include angiotensin II infusion by osmotic minipumps implanted subcutaneously [75], Dahl salt-sensitive rats fed with a high-salt diet [76], and spontaneously hypersensitive rats [77]. These models produce a reliable phenotype of slow-developing cardiac hypertrophy and diastolic dysfunction, which perfectly parallels the pathophysiological progression of HF with preserved ejection fraction in humans. However, the progression to decompensated HF can take up to 25 weeks, where a sudden drop in Dahl sensitive rats survival is noticed [76], or even 12 months for the spontaneously hypersensitive rats [78], increasing maintenance and housing costs when adopting these models.

Several methods have been described to induce RV failure secondary to severe pulmonary hypertension (in primis monocrotaline infusion and sugen/hypoxia models), which have been extensively reviewed by Andersen et al. [79]. Interestingly, angio-proliferative pulmonary hypertension generates more profound fibrotic changes, capillary rarefaction, and contractile impairment in the RV than isolated pressure overload (i.e., from PA banding) [80]. The proposed failure in antioxidant defenses may account for the early development of RV failure 6 weeks after monocrotaline infusion, which differs substantially from the compensated hypertrophic response in the PA banding model, which can be sustained up to 22 weeks [80]. Monocrotaline is proven to induce diffuse sclerosis in other organs, leading to hepatic sinusoidal obstruction [81], renal fibrosis [82], and cachexia [83], which must be taken into account when considering the most appropriate model for HF investigation. The intense systemic oxidative stress and cytokine dysregulation do not represent the usual pathogenic pathways involved in human HF (except for specific myocarditis-based conditions).

Injection of the vascular endothelial growth factor receptor (VEGFR) antagonist SU5416 (known as sugen), followed by a 2–3 week exposure to hypoxia, determines a paradoxical angio-obliterative pulmonary hypertension in rats, due to lung endothelial cell apoptosis [84]. Interbreed differences exist in the RV response to the sugen/hypoxia pulmonary hypertension model. Fisher rats display an early maladaptive RV remodeling leading to exitus by 5 weeks, while Sprague Dawley rats can show preserved RV performance and survival even beyond 9 weeks [86]. Proteomics analysis revealed that a significant downregulation of adenylate kinase 1 (ADK1), which translates into inefficient cardiac energetics, might be the molecular substrate for the premature deterioration of RV function in Fisher rats [86]. In Sprague Dawley rats, RV hypertrophy and increased filling pressures are present after 5 weeks, with a subsequent RV enlargement by 8 weeks, but still preserved RV stroke volume and ejection fraction [85]. Rodent hypoxia-based pulmonary hypertension models have been exhaustively addressed by several reviews [79,84,138].

## 6. Volume Overload Models

Both extracardiac and intracardiac surgical methods have been described to induce volume overload in rats leading to HF (Table 1). The extracardiac approach is the most frequently adopted and entails the creation of an aorto-caval fistula by puncturing the aorta with an 18G needle, which is advanced until perforating the adjacent inferior vena cava. The result is an arterial-venous shunt which creates a high cardiac output HF with volume overload and increased filling pressures into the right chambers. Compensated biventricular hypertrophy is noticed 8 weeks after the procedure, which progresses towards ventricular dilatation in the following weeks [88]. At 12 weeks, pro-arrhythmogenic electrophysiological changes are noticed in the right atrium [89], as well as an increase in LV myocardial stiffness, which is a reliable predictor of animal mortality [90]. After 24 weeks, biventricular failure is established, with reduced contractile function and severe dilatation of both ventricles, although more profoundly in the RV, as reflected by a predominant upregulation of stress and metabolic markers in the RV [87]. Concomitantly, an escalation in animal mortality is evident, whose median survival after the aorto-caval fistula is 43 weeks (in Wistar rats) [87]. The main drawbacks of this technique are the resulting high cardiac output profile, which is a rare cause of HF in the clinical setting, and the artificial mixing of oxygenated and venous blood, which is seen only in HF secondary to congenital heart defects with left-to-right shunt.

Closure of the aorto-caval fistula can be accomplished through a re-laparotomy using a hemoclip [91]. Adopting this temporary volume overloaded model, the reversibility of echocardiographic, histological, and metabolomic changes has been investigated, revealing that, once LV contractile function is compromised (16 weeks after the fistula creation), the correction of the volume overload does not revert LV systolic impairment or repair the defective glycolysis and fatty acid oxidation metabolic pathways [91]. Important considerations from this model might be translated into the clinical decision making for the treatment of aortic and mitral valve regurgitation.

Intracardiac surgical methods include damaging the mitral, pulmonary, or aortic leaflets to generate acute valvular insufficiency and volume overload. The mitral valve is accessed through the LV apex, where a purse-string suture is placed and a 23G needle is advanced under echocardiographic guidance, puncturing the anterior mitral leaflet. After 2 weeks, a regurgitant fraction of 40% is reached, which determinates a progressive enlargement of the LV chamber (end-diastolic volume +28% at 2 weeks, +65% at 10 weeks, +87% at 20 weeks, and +100% at 40 weeks) [92]. A drop in ejection fraction is noticed at 14 weeks with myocyte and extracellular matrix remodeling and a transcriptomic shift illustrating upregulation of oxidative stress pathways [92]. Cardiomyocyte elongation, cytoskeletal disorganization, and loss of mitochondrial networking represent the ultrastructural bases of this remodeling [93]. The low-pressure volume overloaded mitral regurgitation model can be combined with the LAD ligation-based ischemic model to better reproduce the chronic ischemic DCM profile [94]. The additional volume overload accelerates LV dilatation, confirming that mitral regurgitation is a potent driver of adverse cardiac remodeling after a myocardial infarction [95].

Aortic regurgitation is obtained by puncturing the valve with a fixed-core guidewire inserted from the surgically exposed right carotid artery [96]. The temporal sequence in LV remodeling parallels that seen in the mitral regurgitation model, with initial eccentric hypertrophy and dilatation of left-sided chambers (at 8 weeks), followed by a reduction in fractional shortening (at 12 weeks) [97]. Different from the PA banding model, female rats experience a faster progression to LV spherical dilation and wall thickening with reduced circumferential strain, when compared to males [98].

Only one rat model of pulmonary regurgitation is currently available in the scientific literature. By lacerating the pulmonary leaflets with a 22.5 G needle inserted into the main PA through a purse-string suture, pulmonary regurgitation was achieved by Akazawa et al. [99]. In this model, RV dilatation manifested after only 2 weeks, while progressive RV systolic dysfunction occurred after 4 weeks. Due to negative ventricular–ventricular interactions, RV dilatation contributes to diastolic LV compression and impaired relaxation. Interestingly, cardiomyocyte hypertrophy and myocardial fibrosis affected the RV exclusively [99], underlying differential biological pathways supporting ventricular–ventricular interactions in response to RV volume and pressure overload.

Finally, multivalvular regurgitation involving all four cardiac valves has been produced by long-term pergolide and serotonin administration in rats [100], simulating carcinoid heart disease in advanced neuroendocrine tumors.

## 7. Drug Toxicity Models

Drug administration can induce HF in experimental rat models by direct toxicity of myocardial tissue (doxorubicin, ethanol, and homocysteine), producing myocardial ischemia (isoproterenol), type 1 diabetes mellitus (streptozotocin [139]), or pulmonary hypertension (monocrotaline [79]).

Repeated intraperitoneal doxorubicin injections represent one of the most validated rodent models of HF progressing to DCM [140]. By generating mitochondrial dysfunction and intense oxidative stress [101], doxorubicin promotes swelling and vacuolization of cardiomyocytes, disorganization of myofibrils, and intense interstitial fibrosis [102]. Different administration protocols have been described, although a long-term scheme with intraperitoneal weekly injections for a total of 9 weeks (cumulative dose of 18mg/kg) has been found to be more effective to create LV dilatation and systolic impairment [102]. Dose-dependent cardiotoxicity is present, together with a regionalized effect on myocardial contraction, whose earliest impairment is noticed in the basal LV segments by speckle tracking imaging [103].

While experimental papers focus mainly on doxorubicin’s effects on the LV, clinical experience in cancer survivors demonstrated that anthracycline treatment has a detrimental effect also on right-sided chambers [141]. Recent findings suggest increased free radical production in the RV and the conduction system of doxorubicin-treated rats [104], underlining its biventricular toxicity. Given its molecular mechanisms, extracardiac side-effects of doxorubicin are numerous, including hepatic toxicity [105], nephrotic syndrome [106], and bone marrow suppression [107].

Hyperhomocysteinemia in rats is achieved by specific amino acid-defined control diets. After 6 to 10 weeks of treatment, histopathological analysis reveals biventricular hypertrophy of cardiomyocytes, myocardial fiber disarrangement, perivascular and interstitial fibrosis, and inflammatory infiltrate [108,109]. In vivo findings range from LV hypertrophy with preserved systolic function [108] to LV dilatation and loss of contractile function [109].

Acute intravenous administration of ethanol provokes myocardial dysfunction and hypotension, sustained by an abrupt increase in oxidative stress [142]. Chronic moderate ethanol ingestion (8 months of treatment) results in myocyte loss, LV wall thinning, dilatation, and depressed contractile performance [111], while the RV seems to be spared from ethanol toxicity [110].

Several rodent models of diabetes mellitus are available, recently reviewed by Pandey et al. [143]. These models have contributed to the discovery and validation of many antidiabetic molecules and the investigation of diabetes-related multi-organ complications. The intraperitoneal injection of streptozotocin causes a reproducible chemical ablation of pancreatic beta cells, inducing irreversible diabetes in rats. Early signs of diabetic cardiomyopathy can be documented 2–3 weeks after the injection, such as prolonged contraction and relaxation times in isolated cardiomyocytes and enlarged left atrium and impaired systolic parameters with echocardiography [112,113]. Interestingly, these functional alterations appear before the occurrence of morphological changes in the structure of the myocardium and microvasculature [112]. Subsequently, the activation of pro-apoptotic pathways determines the depletion of contractile myocardium which is replaced by fibrotic tissue [114], resulting in LV systo-diastolic dysfunction [116]. Moreover, chronic hyperglycemia triggers secondary pulmonary hypertension which contributes to RV hypertrophy and late systolic dysfunction (12 weeks from disease induction) [115].

## 8. Autoimmune-Mediated Models

Activation of the inflammatory cascade in response to cardiac injury is a common mechanism of ventricular remodeling in many models of HF. After an insult, damaged cardiomyocytes release molecular signals that trigger an acute inflammatory response, digesting necrotic cell debris and promoting subsequent healing pathways [144]. Monocytes and tissue-resident macrophages then acquire a resolution (anti-inflammatory) profile and catalyze the differentiation of myofibroblasts from quiescent fibroblasts [145,146,147]. Myofibroblasts produce the extracellular matrix components that form the basis of scar formation, although if dysregulated, myofibroblasts may also promote a negative fibrotic remodeling process [148]. The inflammatory system initiates the detrimental stimulus which sustains myocardial damage in the setting of acute myocarditis or chronic inflammatory cardiomyopathy [149].

Acute autoimmune myocarditis has been reproduced in rats by injecting porcine myocardial myosin and generating a cross-reactivity with native cardiomyocytes. Using a two-stage protocol of subcutaneous footpad injection of purified porcine cardiac myosin supplemented with complete Freund adjuvant, cardiomyocyte injury, inflammatory infiltrate, and replacement fibrosis are achieved even 18 days after the completion of the injection protocol [117]. Extensive myocardial damage is sustained by several modalities of cell death (i.e., necroptosis, apoptosis, and autophagy) [118]. After 3 weeks, early signs of LV contractile impairment can be detected using cardiac magnetic resonance tissue tracking [119]. Four weeks after the completion of the protocol, LV dilatation and a drop in ejection fraction are evident with standard echocardiography [117]. Interestingly, the RV seems to be almost spared from foci of late gadolinium enhancement with magnetic resonance imaging, suggesting preferential effect on the LV from the acute myocarditis (inflammatory) process [120].

## 9. Rapid Ventricular Pacing Models

Due to practical reasons, rapid pacing-induced HF models usually involve larger mammals, in whom a pacemaker with a pacing lead in the RV apex is implanted subcutaneously and used to maintain supranormal heart rates for several weeks [150]. Rapid pacing has also been adopted in rats to investigate the intracellular effects of chronic tachycardia. Zhou et al. surgically implanted an electrode on the epicardial surface of the RV apex of rats and used it for rapid pacing (550 bpm) for 4 weeks with an external pacemaker [122]. Tachycardia and rapid pacing promoted rat myocyte apoptotic pathways, increased the intracellular levels of reactive oxygen species [121], and dysregulated calcium signaling, especially at higher pacing rates [123]. These cellular mechanisms translated into the development of an apoptotic-based HF profile, with reduced LV contractile function and increased LV end-diastolic pressure after 4 weeks of rapid pacing [122]. Given the need for extracorporeal devices, rapid pacing rat models might be more suitable for the in vitro study of single myocardial fiber response to chronic electric stimulation than whole animal in vivo experiments.

## 10. Genetic Models

Although murine models of HF induced by specific genetic mutations are widely available, few genetically based rat models have been reported. Greaser et al. identified a rat strain with an autosomal dominant mutation of the gene encoding the RNA binding motif protein 20 (Rbm20), which alters the isoform expression of the sarcomeric protein titin (TTN) [124]. Titin is a cardiac and skeletal muscle protein involved in sarcomere assembly and protection from overstretching. Both heterozygous and homozygous Rbm20-deficient rats exhibit a phenotype of DCM with dilated LV, increased subendocardial fibrosis, but initially preserved contractile function [125]. Starting from 3 months of age, progressive LV wall thinning and a decrease in fractional shortening and cardiac output are observed [126]. As seen in humans with hereditary DCM caused by Rbm20 mutations, fibrosis is accompanied by electrical abnormalities that predispose to arrhythmias and sudden cardiac death, which start at 10 months of age in rats [125]. Interestingly, the correction of the cardiac transcriptional profile via base editing can restore cardiac function and revert the HF process in Rbm20 mutant mice [18].

Given its central role in controlling the mechanical properties of the sarcomere, truncating variants of TTN lead to a wide number of inherited myopathies and cardiac disorders [151], ranging from hypertrophic to dilated ventricular profiles. Several mouse models carrying variants in the TTN gene have been developed to understand the structural role of TTN and the clinical correlates in patients. All of the available animal models have been excellently summarized by Marcello and colleagues in a very recent review exclusively focused on TTN pathophysiology [152].

The clinical phenotype of Duchenne muscular dystrophy includes the development of progressive DCM, the leading cause of death in this population [153]. Using CRISPR/Cas9 genome editing of the dystrophin gene, Sugihara and colleagues recently described the first rat model of Duchenne dystrophy with heart involvement [127]. At 10 months of age, mutant rats developed biventricular systolic impairment and myocardial fibrosis. As in the clinical setting, animals also displayed systemic muscular atrophy, with a non-negligible mortality rate. This model provided a suitable platform for the investigation of the potential therapeutic role of a ketogenic diet with medium-chain triglycerides (although the diet was documented to ameliorate skeletal muscle function [154], it triggered a paradoxical exacerbation of the cardiac disorder [128]).

Ling et al. developed a promising heterozygote knockout rat for the myocardium-specific *Isca1*, a causal gene for multiple mitochondrial dysfunction syndromes with cardiac dysplasia [129]. From 3 months of age, rats manifest an echocardiographic and histopathological phenotype of DCM, characterized by LV wall thinning and dilatation, contractile impairment, myocardial lysis and fibrosis. The subcellular mechanisms of mitochondrial dysfunction syndromes were clearly represented in this model, with swollen mitochondria containing damaged membrane structure and partial absence of crests, together with reduced expression levels of key enzymes for ATP synthesis and iron homeostasis [129,130].

Recently, a novel rat model that relies on chemo–genetic interactions was established by Steinhorn et al., injecting rats with adeno-associated virus type 9 carrying a cardiac-specific recombinant D-amino acid oxidase, which produces hydrogen peroxidase during the conversion of D-amino acids into alpha-keto acids [131]. When rats are fed with D-alanine (the substrate of the enzyme D-amino acid oxidase), the acute generation of intracellular hydrogen peroxidase induces a state of oxidative stress in cardiomyocytes. After 4 weeks of the specific diet, rats present HF with reduced LV contractile function and an enlarged LV. Interestingly, LV thickness is maintained and histopathological analysis confirms the absence of fibrosis (at 4 weeks, fibrosis develops later at 8 weeks) [131,133]. Cardiac transcriptome and metabolome analysis revealed marked alterations in mitochondrial function, cardiac energetics, redox homeostasis, amino acid metabolism, cytoskeletal and extracellular matrix organization, and antioxidant systems pathways [132]. This HF model was noticeably reverted by the administration of an angiotensin II receptor blocker, which normalized the echocardiographic, morphologic, and metabolomic markers of disease [132]. Oxidative stress is a common finding in different etiologies of HF, and this model may facilitate the testing of new molecular targets potentially involved in the process of ventricular remodeling.

Finally, several other genetic-based murine models of hereditary cardiomyopathy have been developed. Homozygous desmin knockout and knockin mice display the typical cardiac involvement (LV thinning, fibrosis, and systolic dysfunction) of autosomal-recessive desminopathies [155]. Progressive DCM sustained by cardiomyocyte nuclear envelope disarrangement (envelopathy) can be produced by a knockin mutation of the *LEDM2* gene in mice [156]. Manipulating the expression of desmosomal genes (especially desmocollin-2 and desmoglein-2) causes the development of severe biventricular cardiomyopathy in mice, triggered by an acute inflammatory process leading to cardiomyocyte necrosis and fibrosis [157]. Previous reviews have specifically addressed these topics [31,43,158].

## 11. Conclusions

Adopting the most representative animal model provides the basis for reliable experimental research that translates to clinical practice. Rats provide a strategic compromise between sufficient body size for several morphological evaluations/ease of interventions and low maintenance costs/short breeding cycles. Understanding the advantages and possible drawbacks of each specific rat model can expedite experiments, improve reproducibility, and strengthen reliability. In the present review, we summarize the main characteristics of the currently available rat models of HF and describe their timelines for the development of ventricular failure and phenotypic features to facilitate the standardization and optimization of future investigations in the field of HF.

## Figures and Tables

**Figure 1 ijms-24-03162-f001:**
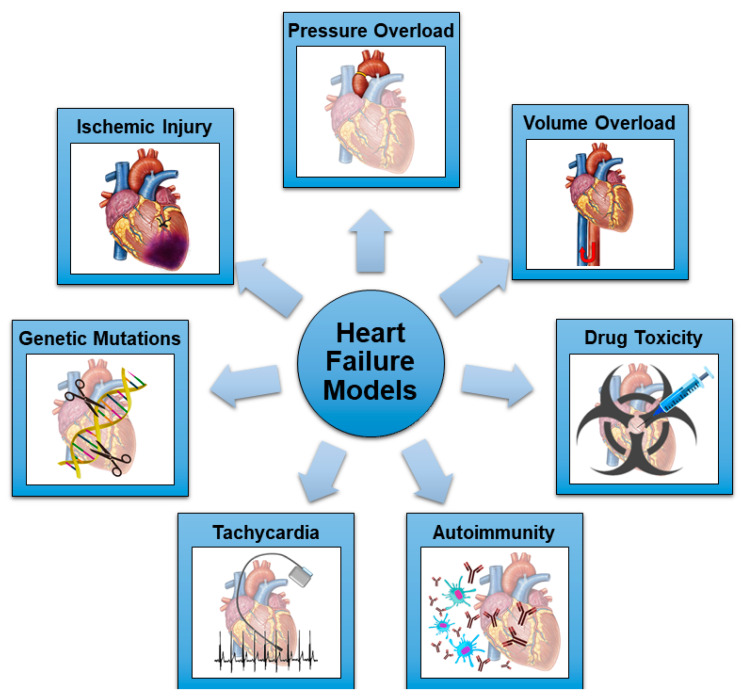
Schematic representation of the most common mechanisms utilized in rodent models of HF.

**Table 1 ijms-24-03162-t001:** Rat Models of HF.

Model	Detrimental Stimulus	Ventricular Selectivity	Advantages	Drawbacks	References
Ischemic Injury	Isoproterenol injections	Predominantly on LV, scarce data on RV	Easiness of administration Validated model Generates reproducible LV dilatation, reduced function, and fibrosis after 2 weeks	Requires daily administrations of isoproterenol for 2 weeks Renal side effects on uric acid excretion [48]	[49]
	Electrocautery injury	LV only, RV only, or biventricular	Selectivity of myocardial damage area Immediate effect of ventricular function No extracardiac side-effects	Requires surgery Absence of a direct correlate with clinical setting	[50]
	Cryogenic damage	LV only, RV only, or biventricular	Selectivity of myocardial damage area Immediate effect of ventricular function No extracardiac side-effects	Requires surgery Absence of a direct correlate with clinical setting	[51]
	Permanent circumflex artery ligation	LV lateral wall	Predictable area of infarction Generates HF after 2–4 weeks No extracardiac side-effects	Less validated Technically more challenging Does not reproduce ischemic-reperfusion injury	[52]
	Permanent LAD ligation	LV anterior or antero-lateral wall	Most validated model Reproducible thanks to standardized surgical protocols Generates HF after 2–4 weeks Presence of a direct correlate with clinical practice No extracardiac side-effects	Requires surgery Scar size varies depending of place of the suture Does not reproduce ischemic-reperfusion injury	[53,54,55,56]
	Temporary LAD/circumflex artery ligation	LV anterior or antero-lateral wall/LV lateral wall	Reproducible thanks to standardized surgical protocols Generates HF after 2–4 weeks Presence of a direct correlate with clinical practice No extracardiac side-effects Replies ischemic-reperfusion injury	Requires surgery Scar size varies depending of place of the suture	[57,58,59]
Pressure Overload	Transverse aortic constriction	Predominantly on LV, scarce data on RV	Reproducible and validated Generates HF after 18 weeks No extracardiac side-effects	Requires surgery Abrupt increase in LV afterload Relatively long time to develop decompensated HF	[60,61] (in mice)
	Ascending aortic banding	Predominantly on LV, RV partially involved by hypertrophy and fibrosis	Reproducible and validated Generates HF after 18 weeks No extracardiac side-effects Mimics the development of HF in severe aortic stenosis	Requires surgery Abrupt increase in LV afterload Relatively long time to develop decompensated HF	[35,62,63,64,65,66,67]
	Abdominal aortic constriction	Predominantly on LV, scarce data on RV	Reproducible Slow-developing HF Useful for studying ventricular remodeling induced by chronic arterial hypertension	Requires surgery Long time to achieve decompensated HF	[68]
	PA banding	Predominantly on RV, scarce data on LV	Reproducible and validated Generates RV failure after 7 weeks No extracardiac side-effects Direct correlation between grade of banding and severity of RV dysfunction	Requires surgery Abrupt increase in RV afterload	[69,70,71,72]
	Temporary aortic banding	Predominantly on LV, RV partially involved by hypertrophy and fibrosis	Reproducible and validated No extracardiac side-effects Useful for investigating ventricular reverse remodelling after aortic valve replacement	Requires two surgeries Debanding is technically challenging Not negligible mortality rate	[63,66,67,73,74]
	Angiotensin II infusion	Predominantly on LV, scarce data on RV	Reproducible and validated Slow-developing HF Useful for studying ventricular remodeling induced by chronic arterial hypertension	Requires a minor surgery for osmotic mini-pump implantation Long time to achieve decompensated HF	[75]
	Dahl sensitive rats on high salt diet	Predominantly on LV, scarce data on RV	Reproducible and validated Slow-developing HF Useful for studying ventricular remodeling induced by chronic arterial hypertension	Long time to achieve decompensated HF (25 weeks) High cost of breed High maintenance costs	[76]
	Spontaneously hypersensitive rats	Predominantly on LV, scarce data on RV	Reproducible and validated Slow-developing HF Useful for studying ventricular remodeling induced by chronic arterial hypertension	Long time to achieve decompensated HF (12 months) High cost of breed High maintenance costs	[77,78]
	Monocrotaline infusion	Predominantly on RV, scarce data on LV	Reproducible and validated Generates RV failure after 6 weeks Mimics RV failure secondary to pulmonary hypertension in humans	Systemic side effects Intense systemic oxidative stress	[79,80,81,82,83]
	Sugen hypoxia	Predominantly on RV	Reproducible and validated Generates fata RV failure by 5 weeks in Fisher rats Progressive transition from RV hypertrophy (5 weeks) to decompensation (>9 weeks) in Sprague-Dawley rats Mimics RV failure secondary to pulmonary hypertension in humans	Interbreed differences Long time to achieve decompensated HF (>9 weeks) in Sprague-Dawley rats	[84,85,86]
Volume Overload	Aorto-caval fistula	Biventricular involvement	Reproducible and validated Generates HF with systolic impairment after 24 weeks No extracardiac side-effects Useful for the investigation of the transition from eccentric hypertrophy to decompensated HF	Requires surgery Long time to achieve decompensated HF High cardiac output profile Mixing of oxygenated and venous blood	[87,88,89,90]
	Temporary aorto-caval fistula	Biventricular involvement	Reproducible No extracardiac side-effects Useful for determining the optimal timing of an effective correction of volume overload	Less validated Technically more challenging Requires two surgeries	[91]
	Mitral regurgitation	Predominantly on LV, scarce data on RV	Reproducible and validated Generates HF with systolic impairment after 14 weeks No extracardiac side-effects Useful for the investigation of the transition from eccentric hypertrophy to decompensated HF Direct correlate with clinical setting	Requires surgery Requires intraoperative echocardiography to target the mitral valve leaflets Relatively long time to achieve decompensated HF	[92,93,94,95]
	Aortic regurgitation	Predominantly on LV, scarce data on RV	Reproducible Generates HF with systolic impairment after 8 weeks No extracardiac side-effects Useful for the investigation of the transition from eccentric hypertrophy to decompensated DCM Does not require thoracotomy Direct correlate with clinical setting	Relatively long time to achieve decompensated HF Sex-related differences in developing HF	[96,97,98]
	Pulmonary regurgitation	Predominantly on RV, diastolic dysfunction in LV	Early progression to RV failure (4 weeks) No extracardiac side-effects Useful for the investigation of ventricular-ventricular interactions	Requires surgery Less validated Technically challenging	[99]
	Pergolide/serotonin injection	Biventricular involvement	Does not require surgery Biventricular involvement Direct correlate with clinical setting (carcinoid heart disease)	Less validated Wide spectrum of possible valvular lesions Poorly predictable phenotype Long time to achieve HF Unknown extracardiac side-effects	[100]
Drug Toxicity	Doxorubicin	Predominantly on LV, scarce data on RV	Reproducible and validated Generates HF after 9 weeks (even before with short-term protocols of injections) Does not require surgery Easiness of administration	Several extra-cardiac side-effects (liver, kidney, and bone marrow toxicity) Intense systemic oxidative stress	[101,102,103,104,105,106,107]
	Homocysteine	Biventricular involvement	Generates HF after 6–10 weeks Does not require surgery Easiness of administration (diet) Direct correlate with clinical setting	Less validated Less reproducible phenotype	[108,109]
	Ethanol	LV only	Does not require surgery Easiness of administration (diet)	Less validated Long time to achieve HF (8 months)	[110,111]
	Streptozotocin	Biventricular involvement	Reproducible and validated Generates early systolic changes in the LV (2 weeks) Does not require surgery Easiness of administration Direct correlate with diabetic cardiomyopathy in humans	Long time to achieve HF (12-20 weeks) Extra-cardiac complications of diabetes mellitus	[112,113,114,115,116]
Auto-immunity	Porcine Myosin Injections	Predominantly on LV	Reproducible Short time to achieve HF (4 weeks) Does not require surgery Easiness of administration Direct correlate with acute myocarditis in clinical setting	Less validated	[117,118,119,120]
Tachy-cardia	Rapid ventricular pacing	Predominantly on LV, no data on RV	Short time to achieve HF (4 weeks)	Poorly validated Poorly standardized Few data on in vivo phenotype of heart failure Need for surgery and extracorporeal pacing More suitable for in vitro applications	[121,122,123]
Genetic Mutations	Rbm20	Predominantly on LV, no data on RV	Reproducible DCM phenotype at 3 months of age Does not require surgery No extracardiac side-effects Direct correlate with the human mutation causing hereditary DCM	Less validated Risk of arrhythmias and sudden death from 10 months of age	[124,125,126]
	Dystrophin	Biventricular involvement	DCM phenotype at 10 months of age Does not require surgery Direct correlate with Duchenne muscular dystrophy	Less validated Extra-cardiac manifestations of muscular atrophy leading to premature animal death	[127,128]
	*Isca1*	Predominantly on LV, no data on RV	Reproducible DCM phenotype at 3 months of age Does not require surgery No extracardiac side-effects Direct correlate with the human mutation causing multiple mitochondrial dysfunction syndromes	Less validated Animal death at 12 months of age	[129,130]
	Chemogenetic (D-amino acid oxidase + D-alanine)	Predominantly on LV, no data on RV	Reproducible Rapid HF phenotype at 4 weeks Does not require surgery No extracardiac side-effects	Less validated	[131,132,133]

DCM: dilated cardiomyopathy; HF: heart failure; LAD: left anterior descending; LV: left ventricle; RV: right ventricle.

## Data Availability

Not applicable.

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
