# Peer review of "Rodent Models of Dilated Cardiomyopathy and Heart Failure for Translational Investigations and Therapeutic Discovery"

_ijms, 2023, doi:10.3390/ijms24043162_

Round 1
Reviewer 1 Report
In the review article 'Rodent models of dilated cardiomyopathy and heart failure for translational investigations and therapeutic discorery' submitted by Ponzoni et al. to the Interantional Journal of Molecular Sciences, the authors summarize the knowledge about rodent models used for DCM.
The topic of this review article is of broad interest. However, the there are some points, which are ignored in this review article:
1.) Please prepare definitions of DCM and compare DCM with other cardiomyopathies like ARVC, HCM or RCM.
2.) Please include genome editing technologies as therapeutic options. The authors should read and include important studies for example on DMD, LEMD2 or RBM20 (base pair editing).
3.) Please update the genetic sequections. Please list more genes and relevant animal models. For example DES, DSC2, TTN and others are completly ignored.
4.) Please cite review articles summarizing animal models for example for other cardiomyopathies like ARVC and HCM (e.g. Gerull and Brodehl, 2020, Frontiers in Physiology, Front Physiol
doi: 10.3389/fphys.2020.00624.).
5.) Inflammation and development of cardiac fibrosis should be discussed in more detail (e.g. Brodehl et al. 2019, PLOSone, Transgenic mice overexpressing desmocollin-2 (DSC2) develop cardiomyopathy associated with myocardial inflammation and fibrotic remodeling).
However, the topic of this review article is really interesting and I am convinced that this review article can be extended in a major revision.
Good luck with the revision!
Author Response
We thank the reviewer for their comments to improve our manuscript. We have provided a summary of our revisions below, we hope that these changes address the concerns of the reviewer.
Comment 1: Please prepare definitions of DCM and compare DCM with other cardiomyopathies like ARVC, HCM or RCM.
Reply: We have added the definition of DCM and we clarified its key features that differentiate it from other cardiomyopathies, as suggested. New references have been added.
Change: “Human DCM is typically defined as a spectrum of myocardial diseases which share ventricular dilatation and depressed contractility[32]. The key phenotype is characterized by a progressive LV dilatation, together with a ventricular shape transition from its original ellipsoid shape to a more spherical one, wall thinning, and a global reduction in contractility, which is revealed by a decrease in stroke volume, cardiac index, and increased strain parameters[30,33]. These features differentiate DCM from other cardiomyopathies, such as hypertrophic cardiomyopathy (where increased LV wall thickness and normal or even supra-normal contractility is noted[34]), restrictive cardiomyopathy (in which ventricular chamber dimensions are reduced, impairing LV filling and creating a primary diastolic dysfunction[35]), and arrhythmogenic right ventricular cardiomyopathy (characterized by typical electrocardiographic anomalies and an often pathognomonic fibrous-fatty myocardial replacement[36])”.
Comment 2: Please include genome editing technologies as therapeutic options. The authors should read and include important studies for example on DMD, LEMD2 or RBM20 (base pair editing).
Reply: We thank the reviewer for this stimulating comment. As indicated, genome editing technologies have been included as therapeutic options for specific forms of dilated cardiomyopathies.
Change: “Genome editing technologies are emerging as potential therapeutic strategies to address specific causative monogenetic disorders associated with the development of cardiomyopathy and HF. Through base editing, the expression of key proteins that are dysregulated in rare genetic forms of dilated cardiomyopathies (DCM), such as dystrophin in Duchenne muscular dystrophy[17] and RBM20[18], can be restored, representing a promising therapeutic concept.”
“Interestingly, the correction of the cardiac transcriptional profile via base editing can restore cardiac function and revert the HF process in RBM20 mutant mice[18].”
Comment 3: Please update the genetic sequections. Please list more genes and relevant animal models. For example, DES, DSC2, TTN and others are completely ignored.
Reply: As suggested, we enriched the chapter regarding the genetic models of HF including titin, dystrophin, desmin, LEMD2, and DSC2 variant models. We provided an in-depth description of available rat models. Given the significant number of exclusively murine genetic models, we referred the reader to recent targeted review papers.
Change: “Given its central role in controlling the mechanical properties of the sarcomere, truncating variants of TTN lead to a wide number of inherited myopathies and cardiac disorders [137], ranging from hypertrophic to dilated ventricular profiles. Several mouse models carrying variants in the TTN gene have been developed to understand the structural role of TTN and the clinical correlates in patients. All of the available animal models have been excellently summarized by Marcello and colleagues in a very recent review exclusively focused on TTN pathophysiology[138].”
“The clinical phenotype of Duchenne muscular dystrophy includes the development of progressive DCM, the leading cause of death in this population[145]. Using CRISPR/Cas9 genome editing of the dystrophin gene, Sugihara and colleagues recently described the first rat model of Duchenne dystrophy with heart involvement[146]. At 10 months of age, mutant rats developed biventricular systolic impairment and myocardial fibrosis. As in the clinical setting, animals also displayed systemic muscular atrophy, with a non-negligible mortality rate. This model provided a suitable platform for the investigation of the potential therapeutic role of a ketogenic diet with medium-chain triglycerides (although the diet was documented to ameliorate skeletal muscle function[147], it triggered a paradoxical exacerbation of the cardiac disorder[148]).”
“Finally, several other genetic-based murine models of hereditary cardiomyopathy have been developed. Homozygous desmin knockout and knockin mice display the typical cardiac involvement (LV thinning, fibrosis, and systolic dysfunction) of autosomal-recessive desminopathies[149]. Progressive DCM sustained by cardiomyocyte nuclear envelope disarrangement (envelopathy) can be produced by a knockin mutation of the LEDM2 gene in mice[150]. Manipulating the expression of desmosomal genes (especially desmocollin-2 and desmoglein-2) causes the development of severe biventricular cardiomyopathy in mice, triggered by an acute inflammatory process leading to cardiomyocyte necrosis and fibrosis[151]. Previous reviews have specifically addressed these topics[31,43,152].”
Comment 4: Please cite review articles summarizing animal models for example for other cardiomyopathies like ARVC and HCM (e.g. Gerull and Brodehl, 2020, Frontiers in Physiology, Front Physiol. 2020 Jun 24;11:624. doi: 10.3389/fphys.2020.00624.).
Reply: We provided additional references concerning animal models of cardiomyopathies other than DCM, which the reader can refer to for a more detailed and comprehensive overview.
Change: “These features differentiate DCM from other cardiomyopathies, such as hypertrophic cardiomyopathy, restrictive cardiomyopathy, and arrhythmogenic right ventricular cardiomyopathy. The available animal models of hypertrophic, restrictive, and arrhythmogenic cardiomyopathies have been extensively reviewed in previous publications[39–44].”
Comment 5: Inflammation and development of cardiac fibrosis should be discussed in more detail (e.g. Brodehl et al. 2019, PLOSone, Transgenic mice overexpressing desmocollin-2 (DSC2) develop cardiomyopathy associated with myocardial inflammation and fibrotic remodeling).
Reply: We thank the reviewer for this suggestion. The inflammatory pathways involved in myocardial healing and cardiac fibrosis have been discussed and the recommended reference has been included in the chapter of genetic HF models.
Change: “Activation of the inflammatory cascade in response to cardiac injury is a common mechanism of ventricular remodeling in many models of HF. After an insult, damaged cardiomyocytes release molecular signals that trigger an acute inflammatory response, digesting necrotic cell debris and promoting subsequent healing pathways[144]. Monocytes and tissue-resident macrophages then acquire a resolution (anti-inflammatory) profile and catalyze the differentiation of myofibroblasts from quiescent fibroblasts[145–147]. Myofibroblasts produce the extracellular matrix components that form the basis of scar formation. Although, if dysregulated, myofibroblasts may also promote a negative fibrotic remodeling process[148]. The inflammatory system can initiate the detrimental stimulus which sustains myocardial damage in the setting of acute myocarditis or chronic inflammatory cardiomyopathy[149].”
“Manipulating the expression of desmosomal genes (especially desmocol-lin-2 and desmoglein-2) promotes the development of severe biventricular cardiomyopathy in mice, triggered by an acute inflammatory process leading to cardiomyocyte necrosis and fibrosis[157].”
Reviewer 2 Report
ijms-2174392, Rodent models of dilated cardiomyopathy and heart failure for translational investigations and therapeutic discovery by Matteo Ponzoni et al. The authors aimed in their review to summarize the main characteristics of the currently available rat models of heart failure (HF), to facilitate the standardization and optimization of future investigations in the field of HF.
The current review article is well structured with concise writing. Sufficient information about previous study findings is presented for readers to follow the present study rationale. Overall, the review has implications for the viability of rodent models of HF that replicate the distinctive features of the major causes of HF in humans. The hope is that HF therapeutic targets identified and tested in animal models, will have a higher likelihood of translating to HF patients
The reviewer has no outstanding concerns but two minor comments:
- Table 1: The reviewer suggests listing all the abbreviations alphabetically in the table footnote.
- Page 11, line 291: Can the authors spell out the abbreviation “VEGFR”?
Author Response
We thank the reviewer for the comments.
Comment 1: The current review article is well structured with concise writing. Sufficient information about previous study findings is presented for readers to follow the present study rationale. Overall, the review has implications for the viability of rodent models of HF that replicate the distinctive features of the major causes of HF in humans. The hope is that HF therapeutic targets identified and tested in animal models, will have a higher likelihood of translating to HF patients.
Reply: We thank the reviewer for this comment and the appreciation of our work.
Change: None needed.
Comment 2: Table 1: The reviewer suggests listing all the abbreviations alphabetically in the table footnote.
Reply: A footnote containing all the abbreviations used in the table was added.
Change: “DCM: dilated cardiomyopathy; HF: heart failure; LAD: left anterior descending; LV: left ventricle; RV: right ventricle.”
Comment 3: Page 11, line 291: Can the authors spell out the abbreviation “VEGFR”?
Reply: The abbreviation of VEGFR was added.
Change: “vascular endothelial growth factor receptor (VEGFR)”.
Round 2
Reviewer 1 Report
The authors significantly improved the manuscript and answered on all points, raised during my first revision. I think the manuscript can be published. Congratulations!